# Frequency of Germline and Somatic *BRCA1* and *BRCA2* Mutations in Prostate Cancer: An Updated Systematic Review and Meta-Analysis

**DOI:** 10.3390/cancers15092435

**Published:** 2023-04-24

**Authors:** Anna Amela Valsecchi, Rossana Dionisio, Olimpia Panepinto, Jessica Paparo, Andrea Palicelli, Francesca Vignani, Massimo Di Maio

**Affiliations:** 1Department of Oncology, University of Turin, Ordine Mauriziano Hospital, 10128 Turin, Italy; annaamela.valsecchi@unito.it (A.A.V.); rdionisio@mauriziano.it (R.D.); olimpia.panepinto@unito.it (O.P.); jessica.paparo@unito.it (J.P.); fvignani@mauriziano.it (F.V.); 2Pathology Unit, Azienda USL-IRCCS di Reggio Emilia, 42123 Reggio Emilia, Italy; andrea.palicelli@ausl.re.it

**Keywords:** prostate cancer, *BRCA1*, *BRCA2*, germline mutation, somatic mutation, meta-analysis

## Abstract

**Simple Summary:**

Our updated systematic review and meta-analysis investigates the frequency of germline and somatic *BRCA1* and *BRCA2* mutations in patients with prostate cancer (PC), with subgroup analysis according to the type of mutation (germline or somatic mutations; mutation of *BRCA1* and/or *BRCA2*) and according to the disease setting (any stage PC or metastatic PC or metastatic castration-resistant PC). As known, *BRCA* testing has recently become standard in clinical practice in prostate cancer because of new available target therapies. However, several open questions remain, in terms of the best time to perform it, the genes to look for (*BRCA* only or genes related to the DNA repair pathway of homologous recombination as well), and the optimal molecular analysis technique (somatic and/or germline testing or, in the future, liquid biopsy, which interestingly could assess both somatic and germline mutations simultaneously).

**Abstract:**

In prostate cancer (PC), the presence of *BRCA* somatic and/or germline mutation provides prognostic and predictive information. Meta-analysis aims to estimate the frequency of *BRCA* mutations in patients with PC (PCp). In November 2022, we reviewed literature searching for all articles testing the proportion of *BRCA* mutations in PCp, without explicit enrichment for familiar risk. The frequency of germline and somatic *BRCA1* and/or *BRCA2* mutations was described in three stage disease populations (any/metastatic/metastatic castration-resistant PC, mCRPC). Out of 2253 identified articles, 40 were eligible. Here, 0.73% and 1.20% of any stage PCp, 0.94% and 1.10% of metastatic PCp, and 1.21% and 1.10% of mCRPC patients carried germline and somatic BRCA1 mutation, respectively; 3.25% and 6.29% of any stage PCp, 4.51% and 10.26% of metastatic PCp, and 3.90% and 10.52% of mCRPC patients carried germline and somatic BRCA2 mutation, respectively; and 4.47% and 7.18% of any stage PCp, 5.84% and 10.94% of metastatic PCp, and 5.26% and 11.26% of mCRPC patients carried germline and somatic *BRCA1/2* mutation, respectively. Somatic mutations are more common than germline and *BRCA2* are more common than *BRCA1* mutations; the frequency of mutations is higher in the metastatic setting. Despite that *BRCA* testing in PC is now standard in clinical practice, several open questions remain.

## 1. Introduction

In oncology, the demand for breast cancer gene (*BRCA*) genetic testing in various tumor types, such as ovarian, breast, pancreatic, and prostate cancer (PC), is rapidly and continuously increasing to predict the efficacy of cancer treatment, help physicians make decisions about therapeutic options, and assess individual and familial risk [1,2].

Regarding patients with PC, knowledge of the presence of *BRCA1/2* mutations in cancer tissue (somatic mutations) or in peripheral blood (germline mutations) provides useful information of prognostic and predictive value.

In particular, first, *BRCA* mutation identification allows the planning of an appropriate therapeutic algorithm. Indeed, *BRCA* testing is essential to determine whether patients are eligible for new targeted and effective therapeutic strategies, such as poly-ADP-ribose polymerase inhibitors (PARPis). While treatment of metastatic PC has historically consisted of hormonal therapy with androgen deprivation, chemotherapy, and various radiotherapy approaches, the recent approval of PARPis, such as rucaparib and olaparib, has revolutionized the therapeutic algorithm of metastatic castration-resistant PC (mCRPC) and led to a marked improvement in clinical outcomes for patients with *BRCA1/2* mutations [3,4,5,6,7].

Second, the identification of a pathogenetic germline variant in *BRCA* genes provides access to prevention programs, oncogenetic counseling of family members to identify high-risk carriers, special screening programs for early detection of *BRCA*-related heredo-familial tumors, and risk-reduction strategies [8].

*BRCA* testing requires standardized and harmonized procedures for germline and tumor DNA sequencing and for the interpretation of results; *BRCA* mutational status should be verified by a specialized laboratory using a validated analytical method [9,10].

According to the latest position paper of Italian Scientific Societies and the most recent European Society of Medical Oncology (ESMO) clinical practice guidelines, it is preferable to investigate pathogenetic *BRCA* variants in tumor tissue first, as the probability of detecting *BRCA* mutations is higher than with germline analysis. Patients who are found to have somatic pathogenic *BRCA* mutations should be referred for germline testing to identify possible constitutional and hereditary variants. Somatic testing should also be proposed to patients who initially underwent germline testing that did not identify a pathogenic variant and who are potential candidates for treatment with PARPis [9,10].

Data on the exact proportion of PC patients with *BRCA* mutations come from a 2018 systematic review and meta-analysis by Mok et al. They showed that the frequency of *BRCA1* and *BRCA2* carriers in PC patients was 0.9% and 2.2%, respectively [11].

As these data did not include more recent studies, and *BRCA* testing is now standard in clinical practice in metastatic PC thanks to the approval of specific treatments, we decided to conduct an updated systematic literature review and meta-analysis with the aim of evaluating the proportion of PC patients with *BRCA* mutations, dividing the data obtained into subgroups according to the type of mutation (germline or somatic mutations; mutation of *BRCA1* and/or *BRCA2*) and according to the disease setting (any stage or metastatic PC or mCRPC).

## 2. Materials and Methods

This systematic review was conducted according to the Preferred Reporting Items for Systematic Review and meta-analysis (PRISMA) guidelines, as reported in Figure 1.

### 2.1. Search Strategy

An extensive literature search in PubMed, Web of Sciences, and Scopus databases was performed in November 2022 to identify all articles testing the proportion of BRCA1 and BRCA2 mutations in patients with PC.

The following keywords were used in our search strategy: “(prostate cancer) and (BRCA)”, “(prostate cancer) and (BRCA1 gene)”, “(prostate cancer) and (BRCA2 gene)”, “(prostate cancer) and (BRCA mutation)”, “(prostate cancer) and (BRCA testing)”, “(prostate cancer) and (germline BRCA)”, and “(prostate cancer) and (somatic BRCA)”. References of the identified articles were also checked manually to identify additional eligible items.

Initial screening was performed by one investigator (A.A.V.) and ineligible results were identified based on the titles and abstracts. If the study’s topic could not be ascertained from its title or abstract, the full-text version would be retrieved for evaluation. Disagreement was resolved by discussion or consensus with another co-author (M.D.M.).

### 2.2. Study Selection

To have sufficient data to calculate the number of *BRCA* mutation carriers among patients with PC, studies were screened for eligibility using the following inclusion criteria: (1) participants must be patients with PC, regardless of disease stage; (2) included studies must report the proportion of patients with *BRCA* mutations tested by somatic and/or germline testing, regardless of the gene involved (*BRCA1* or *BRCA2* or any *BRCA*) and mutation variant; and (3) articles must be in English and published between 2000 and 2022.

The following criteria were used as exclusion criteria: (1) participants with established risk factors for PC such as patients with inherited PC or patients with relatives with PC and (2) case reports and reviews.

### 2.3. Data Collection

For each eligible article, the following data were collected: (1) first author’s name; (2) year of publication; (3) total number of patients; (4) number of patients with or without BRCA mutations; (5) details of population disease setting: any stage PC, metastatic PC, and mCRPC; and (6) details of type of *BRCA* mutation: germline, somatic, *BRCA1*, and *BRCA2*.

### 2.4. Statistical Methods

The meta-analysis of the proportion of patients with PC with *BRCA* mutations was performed with MedCalc Statistical Software version 20.211 (MedCalc Software Ltd., Ostend, Belgium; https://www.medcalc.org; 2023). The software uses a Freeman–Tukey transformation (arcsine square root transformation) to calculate the weighted summary proportion under the fixed and random effects model. Heterogeneity is measured by Cochran’s Q, calculated as the weighted sum of squared differences between the individual study proportion and the pooled proportion across studies. Q is distributed as a chi-square statistic with k (number of studies) minus 1 degrees of freedom. The I^2^ statistic describes the percentage of variation across studies that is due to heterogeneity rather than chance. I^2^ = 100% × (Q − df)/Q.

### 2.5. Role of Funding Source

There was no funding source for this systematic review and meta-analysis. All authors had full access to all data and the corresponding author (M.D.M.) had the final responsibility for the decision to submit for publication.

## 3. Results

Our research items led to the identification of 2253 titles. After removing duplicates, non-pertinent items and ineligible studies, 40 articles were included in this systematic review and meta-analysis (Figure 1; Table 1) [12,13,14,15,16,17,18,19,20,21,22,23,24,25,26,27,28,29,30,31,32,33,34,35,36,37,38,39,40,41,42,43,44,45,46,47,48,49,50,51].

See Appendix A for all detailed statistical results of the meta-analysis.

### 3.1. Meta-Analysis: Proportion of Patients with Prostate Cancer with BRCA1 Mutation

#### 3.1.1. Proportion of Patients with Any Stage PC with BRCA1 Mutation

The proportion of germline BRCA1 mutation carriers among patients with any stage PC was available from 31 articles, for a total of 32,525 patients, and was equal to 0.73% (95% confidence interval, CI: 0.51–1.00), with significant heterogeneity (I^2^ = 81.19%; *p* < 0.0001) (Figure 2a).

The proportion of somatic BRCA1 mutation carriers among patients with any stage PC was available from 10 articles, for a total of 3229 patients, and was equal to 1.20% (95% CI: 0.85–1.60), without significant heterogeneity (I^2^ = 0.00%; *p* = 0.7423) (Figure 2b).

#### 3.1.2. Proportion of Patients with Metastatic PC with BRCA1 Mutation

The proportion of germline BRCA1 mutation carriers among patients with metastatic PC was available from 10 articles, for a total of 3963 patients, and was equal to 0.94% (95% CI: 0.19–2.23), with significant heterogeneity (I^2^ = 88.85%; *p* < 0.0001) (Figure 2c).

The proportion of somatic BRCA1 mutation carriers among patients with metastatic PC was available from six articles, for a total of 1384 patients, and was equal to 1.10% (95% CI: 0.62–1.71), without significant heterogeneity (I^2^ = 0.00%; *p* = 0.9224) (Figure 2d).

#### 3.1.3. Proportion of Patients with mCRPC with BRCA1 Mutation

The proportion of germline BRCA1 mutation carriers among patients with mCRPC was available from seven articles, for a total of 2571 patients, and was equal to 1.21% (95% CI: 0.053–3.84), with significant heterogeneity (I^2^ = 92.36%; *p* < 0.0001) (Figure 2e).

The proportion of somatic BRCA1 mutation carriers among patients with mCRPC was available from five articles, for a total of 1243 patients, and was equal to 1.10% (95% CI: 0.60–1.76), without significant heterogeneity (I^2^ = 0.00%; *p* = 0.8425) (Figure 2f).

### 3.2. Meta-Analysis: Proportion of Patients with Prostate Cancer with BRCA2 Mutation

#### 3.2.1. Proportion of Patients with Any Stage PC with BRCA2 Mutation

The proportion of germline BRCA2 mutation carriers among patients with any stage PC was available from 30 articles, for a total of 29,813 patients, and was equal to 3.25% (95% CI: 2.54–4.04), with significant heterogeneity (I^2^ = 90.96%; *p* < 0.0001) (Figure 3a).

The proportion of somatic BRCA2 mutation carriers among patients with any stage PC was available from 10 articles, for a total of 3229 patients, and was equal to 6.29% (95% CI: 3.79–9.38), with significant heterogeneity (I^2^ = 89.14%; *p* < 0.0001) (Figure 3b).

#### 3.2.2. Proportion of Patients with Metastatic PC with BRCA2 Mutation

The proportion of germline BRCA2 mutation carriers among patients with metastatic PC was available from 10 articles, for a total of 3963 patients, and was equal to 4.51% (95% CI: 2.93–6.42), with significant heterogeneity (I^2^ = 81.54%; *p* < 0.0001) (Figure 3c).

The proportion of somatic BRCA2 mutation carriers among patients with metastatic PC was available from six articles, for a total of 1384 patients, and was equal to 10.26% (95% CI: 7.92–12.85), without significant heterogeneity (I^2^ = 38.42%; *p* = 0.1498) (Figure 3d).

#### 3.2.3. Proportion of Patients with mCRPC with BRCA2 Mutation

The proportion of germline BRCA2 mutation carriers among patients with mCRPC was available from seven articles, for a total of 2571 patients, and was equal to 3.90% (95% CI: 2.13–6.16), with significant heterogeneity (I^2^ = 76.71%; *p* = 0.0002) (Figure 3e).

The proportion of somatic BRCA2 mutation carriers among patients with mCRPC was available from five articles, for a total of 1243 patients, and was equal to 10.52% (95% CI: 7.64–13.81), without significant heterogeneity (I^2^ = 49.50%; *p* = 0.0945) (Figure 3f).

### 3.3. Meta-Analysis: Proportion of Patients with Prostate Cancer with Any BRCA Mutation

#### 3.3.1. Proportion of Patients with Any Stage PC with Any BRCA Mutation

The proportion of germline BRCA1/2 mutation carriers among patients with any stage PC was available from 29 articles, for a total of 33,784 patients, and was equal to 4.47% (95% CI: 3.38–5.70), with significant heterogeneity (I^2^ = 95.57%; *p* < 0.0001) (Figure 4a).

The proportion of somatic BRCA1/2 mutation carriers among patients with any stage PC was available from 10 articles, for a total of 3229 patients, and was equal to 7.18% (95% CI: 4.89–9.87), with significant heterogeneity (I^2^ = 84.17%; *p* < 0.0001) (Figure 4b).

#### 3.3.2. Proportion of Patients with Metastatic PC with Any BRCA Mutation

The proportion of germline BRCA1/2 mutation carriers among patients with metastatic PC was available from 11 articles, for a total of 11,670 patients, and was equal to 5.84% (95% CI: 3.72–8.41), with significant heterogeneity (I^2^ = 93.61%; *p* < 0.0001) (Figure 4c).

The proportion of somatic BRCA1/2 mutation carriers among patients with metastatic PC was available from six articles, for a total of 1384 patients, and was equal to 10.94% (95% CI: 8.73–13.36), without significant heterogeneity (I^2^ = 29.07%; *p* = 0.2170) (Figure 4d).

#### 3.3.3. Proportion of Patients with mCRPC with Any BRCA Mutation

The proportion of germline BRCA1/2 mutation carriers among patients with mCRPC was available from seven articles, for a total of 2571 patients, and was equal to 5.26% (95% CI: 2.18–9.57), with significant heterogeneity (I^2^ = 91.57%; *p* < 0.0001) (Figure 4e).

The proportion of somatic BRCA1/2 mutation carriers among patients with mCRPC was available from five articles, for a total of 1243 patients, and was equal to 11.26% (95% CI: 8.49–14.38), without significant heterogeneity (I^2^ = 42.38%; *p* = 0.1390) (Figure 4f).

## 4. Discussion

In this systematic review and meta-analysis, we collected all papers describing the frequency of somatic and/or germline *BRCA1* and *BRCA2* mutations in patients with PC. We analyzed this frequency in three populations of patients: all PC patients regardless of the stage, patients with metastatic PC, and patients with mCRPC.

First, although the complete information about somatic and germline status was available only in a subset of studies, we confirmed that, overall, somatic *BRCA* mutations are markedly more frequent than germline mutations: 7.18% versus 4.47%, respectively, in all patients with PC regardless of the stage of PC disease; 10.94% versus 5.84%, respectively, in patients with metastatic disease; and 11.26% versus 5.26%, respectively, in patients with mCRPC. Data obtained considering mutations of *BRCA1* or *BRCA2* separately confirmed a higher frequency of somatic mutations than germline mutations for both genes.

Second, both germline and somatic *BRCA2* mutations are more common than *BRCA1* mutations in both metastatic and patients with any stage PC. Specifically, among metastatic patients, 10.26% and 4.51% of cases have somatic and germline *BRCA2* mutations, respectively, while 1.1% and 0.94% have somatic and germline *BRCA1* mutations, respectively. Among patients with any stage PC, 6.29% and 3.25% have somatic and germline *BRCA2* mutations, respectively, while 1.20% and 0.73% have somatic and germline *BRCA1* mutations, respectively.

Finally, the frequency of *BRCA* mutations is higher in the series including only patients with metastatic disease than in the whole population of all patients studied, regardless of stage. Namely, the frequency of somatic *BRCA1/2* mutations is 10.94% in patients with metastatic disease (11.26% when the analysis is limited to the castration-resistant setting) and 7.18% in all patients with any stage PC.

Similar to other solid tumors, including breast and ovarian cancer, in prostate cancer, the presence of *BRCA* mutation is an important clinical factor with prognostic and predictive value, especially owing to the recent introduction of target therapies such as PARPis into clinical practice. To date, these drugs have only been approved for mCRPC disease, although several studies are underway to predict their use in earlier stages of PC [52,53,54]. Therefore, molecular characterization of patients with PC is essential to avoid depriving them of a potential effective therapeutic option.

Our data confirm that many more cases can be identified with the somatic test than with the germline test alone. Therefore, the possibility of performing the somatic test must be guaranteed in all oncological centers. Until a few years ago, the only relevant determination in clinical practice was the search for germline mutations, in the context of genetic counseling for known or suspected hereditary cases. Nowadays, with the availability of target drugs, the determination of *BRCA* mutational status becomes relevant for therapeutic choices, and this implies a marked increase in the number of cases eligible for testing, as well as the need to obtain results more quickly in order to allow timely therapeutic decisions. This is a good example of the risk of disparities among different countries and different centers, owing to the asymmetry in reimbursement systems and in technical pathways for carrying out molecular tests; that is, patients could be at risk of unequal access not only to drugs, but also to tests.

In our meta-analysis, we focused on evaluating only the rate of *BRCA1* and *BRCA2* genes. Actually, although the real predictive value of other genes is controversial, mutations in genes related to the DNA repair pathway of homologous recombination (HR) (HRD-positive patients) have also been proposed and studied as predictive factors for PARPis. Therefore, in addition to *BRCA* mutations, other HRD-related gene aberrations may also serve as novel biomarkers for predicting the efficacy of PARPis [55].

However, in PC, the recommendations in the various international guidelines are not entirely congruent.

The 2022 Italian Association of Medical Oncology (AIOM) guidelines recommend *BRCA* testing for all patients with metastatic PC, without a recommendation about other genes. Namely, the indication to perform the test is also extended to patients who meet certain criteria regarding personal and family history, number of affected relatives, cancer type, multiple primary tumors, and age at diagnosis, as well as histologic, immunohistochemical, and molecular tumor characteristics [9].

Instead, European guidelines, issued by ESMO in 2020, recommend that tissue-based molecular assays may be used in conjunction with clinicopathological factors for treatment decisions in localized prostate cancer; germline testing for *BRCA2* and other DDR genes associated with cancer predisposition syndromes is recommended in patients with a family history of cancer and should be considered in all patients with metastatic PC; tumor testing for HR genes can be considered in patients with mCRPC [10].

Still somewhat different are the recommendations of the National Comprehensive Cancer Network (NCCN) guidelines published in 2023: germline multigene testing that includes at least *BRCA1*, *BRCA2*, *ATM*, *PALB2*, *CHEK2*, *HOXB13*, *MLH1*, *MSH2*, *MSH6*, and *PMS2* is recommended if the patient is affected by metastatic, regional (node positive), very-high-risk localized, or high-risk localized PC (diagnosed at any age) and/or if certain criteria about family history and/or ancestry are met, while tumor testing for alterations in HR DNA repair genes, such as *BRCA1*, *BRCA2*, *ATM*, *PALB2*, *FANCA*, *RAD51D*, *CHEK2*, and *CDK12*, is recommended in patients with metastatic PC and can be considered in patients with regional PC [5].

The test has also recently acquired, in addition to the traditional implications for the management of hereditary–familial cases, implications for the therapeutic management of patients. At least in part, probably, this fact explains the heterogeneity between different recommendations and guidelines.

Our meta-analysis, although based on a systematic and updated review of the literature, has some important limitations.

First of all, the absence of individual patient data implies that the information on the characteristics of the patients in the studies is limited. Because the incidence of mutations according to individual characteristics for all patients, their ethnicity, or geographic origin are not uniformly available, clinical characteristics associated with the presence of a mutation could not be analyzed. As for the association with clinical stage, the higher frequency of mutations in patients with metastatic disease seems more relevant for somatic than for germline mutations. However, we are unable to establish the exact timing of the appearance of somatic mutations with respect to disease progression, also because most somatic tests are performed on tissue previously archived at the time of initial diagnosis. Only serial tests on tissue samples taken at different stages of the disease could establish whether, in cases of wildtype for germline mutations, the appearance of somatic mutations is an early event potentially associated with higher risk of metastases and a worse prognosis or simply a late event in the natural history of the disease. However, the execution of molecular testing on archived tissue is consistent with daily clinical practice. Thus, the collected data help to estimate the number of patients with PC with *BRCA* mutations that we can expect to see in clinical practice. In this scenario, it would be important for urologists to be aware at the time of a prostate biopsy that the tissue is not only needed for histologic analysis, but could also be useful for genetic analysis. The biopsy or tissue removed should be quantitatively sufficient for both analyses so that the patient is not biopsied again later.

Another limitation is the heterogeneity that characterizes the techniques of molecular analysis used in the studies included in the meta-analysis. Different techniques can be different for sensitivity and specificity, and this could contribute to the high heterogeneity found in the incidence of mutations among different studies. Furthermore, different *BRCA* mutations variants were not uniformly distinguished according to their predictive value.

Lastly, our meta-analysis did not include studies that investigated *BRCA* mutations using the liquid biopsy technique. According to recent studies, liquid biopsy seems to have a very interesting role for three main reasons.

First, a study by Tukachinsky et al. showed that there is a good agreement between data obtained from somatic testing and those obtained from liquid biopsy [46]. Thus, the liquid biopsy technique would allow to assess both somatic and germline mutations simultaneously using only one blood sample.

Second, a recent exploratory analysis of the PROfound study evaluated the efficacy of olaparib in patients with *BRCA/ATM* mutations investigated by liquid biopsy, showing that the clinical outcome endpoints were similar to those reached in the cohort in which mutations had been studied with somatic testing [3,56]. Therefore, this study highlights that the liquid biopsy technique could be a test with the same prognostic and predictive value as somatic testing.

Third, somatic testing has failure rates for several reasons, such as a lack of quantitatively sufficient tumor tissue or other technical difficulties. For example, in the PROfound study, the success rate of somatic testing was 69% [3]. In this scenario, liquid biopsy could exceed the limitations of somatic testing and become a valid and useful alternative.

Therefore, the role of liquid biopsy will become increasingly intriguing in the future because it could offer our patients with mCRPC a less invasive technique than somatic testing that can overcome its limitations while maintaining the ability to provide predictive and prognostic information.

## 5. Conclusions

In prostate cancer, knowledge of the presence of somatic and/or germline mutations of *BRCA* provides useful information of prognostic and predictive value to plan an appropriate therapeutic algorithm thanks to the introduction of new therapeutic options and ensure access to prevention programs and oncogenetic counseling.

In summary, as *BRCA* testing is now well-established in clinical practice, this meta-analysis aimed to describe the rate of *BRCA* mutations that clinicians should expect to see on a daily basis.

Meta-analysis demonstrates that somatic mutations are more common than germline mutations, *BRCA2* mutations are more common than *BRCA1* mutations in both metastatic patients and patients with any stage PC, and that the frequency of *BRCA* mutations is higher in the series including only patients with metastatic disease than in the whole population of all patients studied regardless of stage.

Because the test has recently acquired implications for the therapeutic management of patients with PC, the recommendations in the various international guidelines are not entirely congruent and, in this scenario, several questions remain for the future, both in terms of the best time to perform *BRCA* testing based on ongoing studies of the use of PARPis at an earlier stage of PC disease, as well as in terms of the genes to look for (*BRCA* or HRD panel) and the optimal molecular analysis technique (somatic and/or germline testing or liquid biopsy).

## Figures and Tables

**Figure 1 cancers-15-02435-f001:**
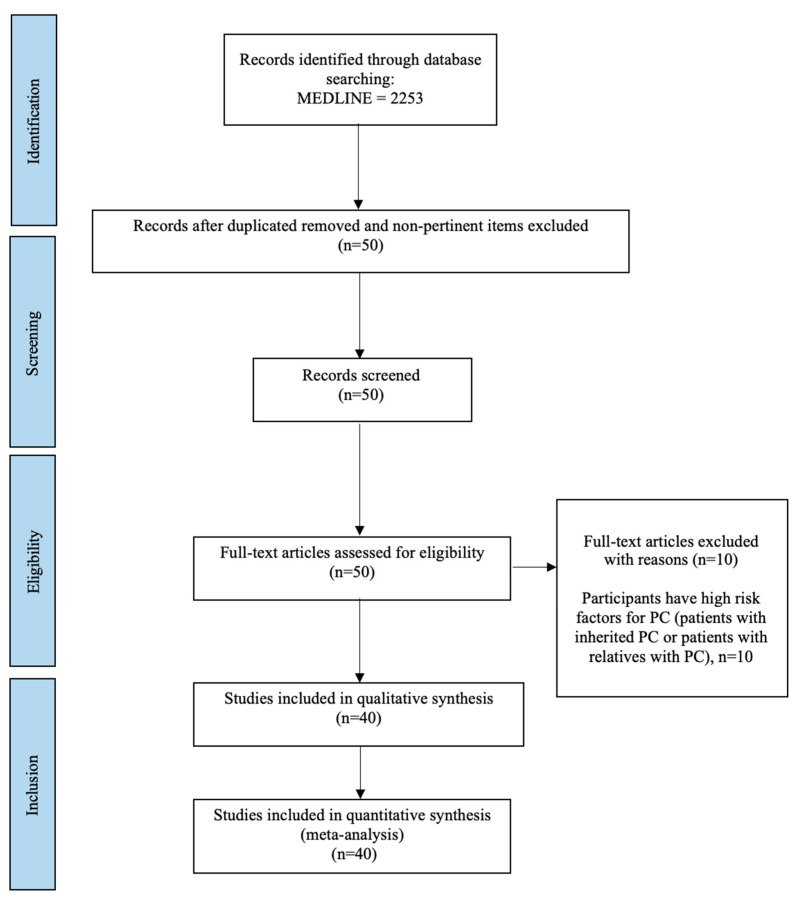
PRISMA diagram.

**Figure 2 cancers-15-02435-f002:**
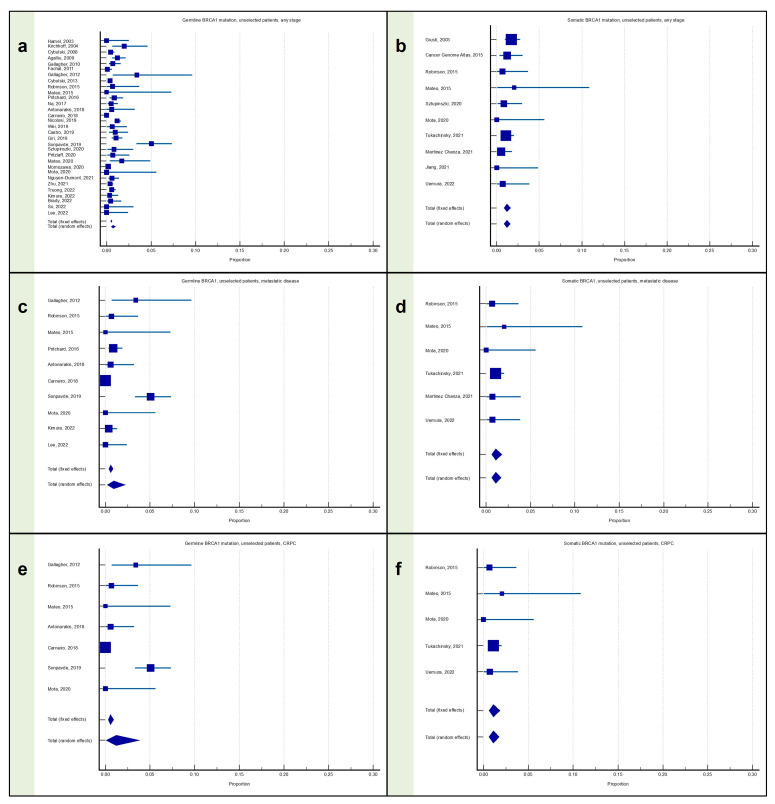
Proportion of patients with prostate cancer harboring the BRCA1 mutation. (**a**) Proportion of patients with any stage PC with the germline BRCA1 mutation; (**b**) proportion of patients with any stage PC with the somatic BRCA1 mutation; (**c**) proportion of patients with metastatic PC with the germline BRCA1 mutation; (**d**) proportion of patients with metastatic PC with the somatic BRCA1 mutation; (**e**) proportion of patients with mCRPC with the germline BRCA1 mutation; and (**f**) proportion of patients with mCRPC with the somatic BRCA1 mutation [12,13,14,15,16,18,19,20,22,23,24,25,26,27,28,29,30,31,32,33,34,37,38,39,40,41,42,44,45,46,47,48,49,50,51].

**Figure 3 cancers-15-02435-f003:**
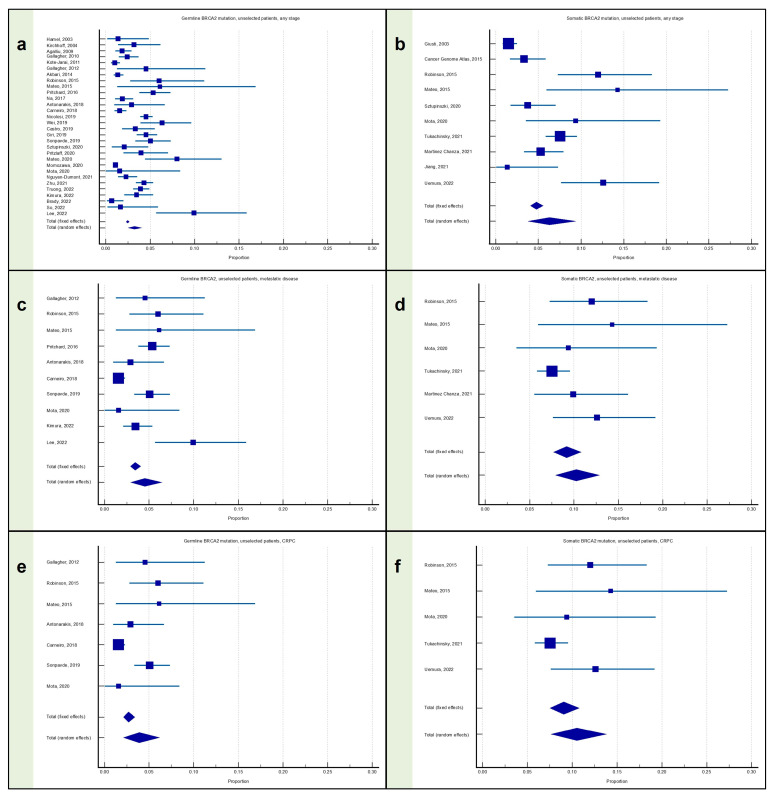
Proportion of patients with prostate cancer harboring the BRCA2 mutation. (**a**) Proportion of patients with any stage PC with the germline BRCA2 mutation; (**b**) proportion of patients with any stage PC with the somatic BRCA2 mutation; (**c**) proportion of patients with metastatic PC with the germline BRCA2 mutation; (**d**) proportion of patients with metastatic PC with the somatic BRCA2 mutation; (**e**) proportion of patients with mCRPC with the germline BRCA2 mutation; and (**f**) proportion of patients with mCRPC with the somatic BRCA2 mutation [12,13,15,16,17,18,19,21,22,23,24,25,26,27,28,29,30,31,32,33,34,35,37,38,39,40,41,42,43,44,45,46,47,48,49,50,51].

**Figure 4 cancers-15-02435-f004:**
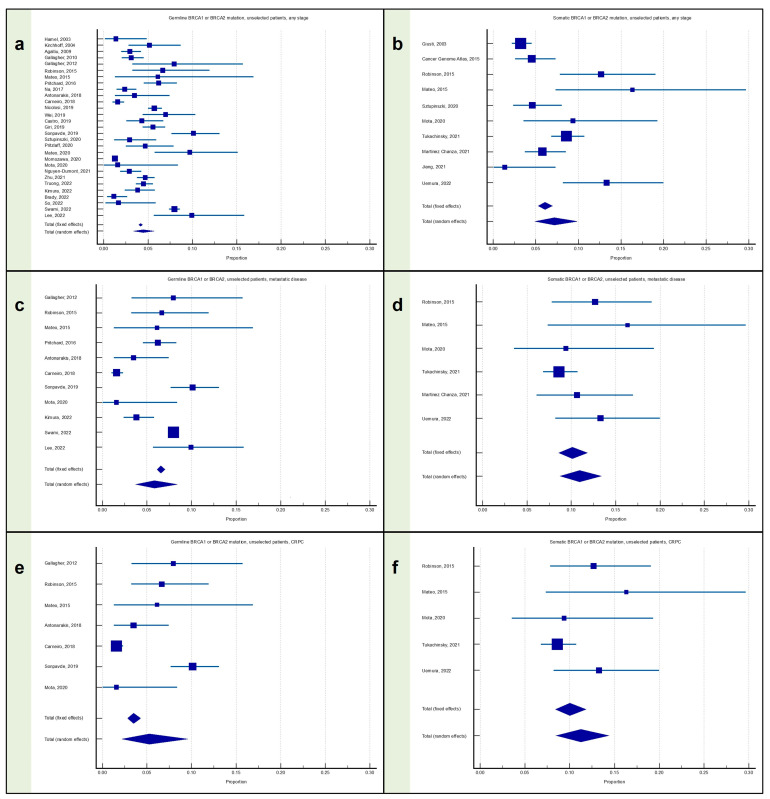
Proportion of patients with prostate cancer harboring any BRCA mutation. (**a**) Proportion of patients with any stage PC with the germline BRCA1/2 mutation; (**b**) proportion of patients with any stage PC with the somatic BRCA1/2 mutation; (**c**) proportion of patients with metastatic PC with the germline BRCA1/2 mutation; (**d**) proportion of patients with metastatic PC with the somatic BRCA1/2 mutation; (**e**) proportion of patients with mCRPC with the germline BRCA1/2 mutation; and (**f**) proportion of patients with mCRPC with the somatic BRCA1/2 mutation [12,13,15,16,19,22,23,24,25,26,27,28,29,30,31,32,33,34,35,36,37,38,39,40,41,42,43,44,45,46,47,48,49,50,51].

**Table 1 cancers-15-02435-t001:** Summary of the results of the meta-analysis.

	*BRCA1*		*BRCA2*		*BRCA1/2*	
	Germline	Somatic	Germline	Somatic	Germline	Somatic
**Patients with any stage PC**						
Number of studies	31	10	30	10	29	10
Number of patients	32,525	3229	29,813	3229	33,784	3229
% (fixed effect)	0.53 (95% CI: 0.45–0.62)	1.20 (95% CI: 0.85–1.64)	2.47 (95% CI: 2.30–2.66)	4.77 (95% CI: 4.06–5.56)	4.17 (95% CI: 3.96–4.39)	6.07 (95% CI: 5.27–6.95)
% (random effect)	0.73 (95% CI: 0.51–1.00)	1.20 (95% CI: 0.85–1.60)	3.25 (95% CI: 2.54–4.04)	6.29 (95% CI: 3.79–9.38)	4.47 (95% CI: 3.38–5.70)	7.18 (95% CI: 4.89–9.87)
Heterogeneity I^2^ (*p*-value)	81.19% (*p* < 0.0001)	0.00% (*p* = 0.7423)	90.96% (*p* < 0.0001)	89.14% (*p* < 0.0001)	95.57% (*p* < 0.0001)	84.17% (*p* < 0.0001)
**Metastatic PC patients**						
Number of studies	10	6	10	6	11	6
Number of patients	3963	1384	3963	1384	11,670	1384
% (fixed effect)	0.58 (95% CI: 0.37–0.87)	1.10 (95% CI: 0.62–1.79)	3.44 (95% CI: 2.89–4.05)	9.16 (95% CI: 7.70–10.80)	6.56 (95% CI: 6.12–7.03)	10.12 (95% CI: 8.58–11.82)
% (random effect)	0.94 (95% CI: 0.19–2.23)	1.10 (95% CI: 0.62–1.71)	4.51 (95% CI: 2.93–6.42)	10.26 (95% CI: 7.92–12.85)	5.84 (95% CI: 3.72–8.41)	10.94 (95% CI: 8.73–13.36)
Heterogeneity I^2^ (*p*-value)	88.85% (*p* < 0.0001)	0.00% (*p* = 0.9224)	81.54% (*p* < 0.0001)	38.42% (*p* = 0.1498)	93.61% (*p* < 0.0001)	29.07% (*p* = 0.2170)
**mCRPC patients**						
Number of studies	7	5	7	5	7	5
Number of patients	2571	1243	2571	1243	2571	1243
% (fixed effect)	0.56 (95% CI: 0.31–0.93)	1.10 (95% CI: 0.60–1.85)	2.69 (95% CI: 2.10–3.39)	9.05 (95% CI: 7.51–10.78)	3.50 (95% CI: 2.82–4.28)	10.03 (95% CI: 8.42–11.83)
% (random effect)	1.21 (95% CI: 0.05–3.84)	1.10 (95% CI: 0.60–1.76)	3.90 (95% CI: 2.13–6.16)	10.52 (95% CI: 7.64–13.81)	5.26 (95% CI: 2.18–9.57)	11.26 (95% CI: 8.49–14.38)
Heterogeneity I^2^ (*p*-value)	92.36% (*p* < 0.0001)	0.00% (*p* = 0.8425)	76.71% (*p* = 0.0002)	49.50% (*p* = 0.0945)	91.57% (*p* < 0.0001)	42.38% (*p* = 0.1390)

PC: prostate cancer; mCRPC: metastatic castration-resistant prostate cancer; CI: confidence interval.

## Data Availability

The data presented in this study are available in Appendix A.

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
