# Peer review of "Frequency of Germline and Somatic BRCA1 and BRCA2 Mutations in Prostate Cancer: An Updated Systematic Review and Meta-Analysis"

_cancers, 2023, doi:10.3390/cancers15092435_

Round 1

Reviewer 1 Report

The authors performed a systematic review and meta-analysis of BRCA gene 1 & 2 genetic alterations in PCa.

In general, the manuscript is well written and timely.

My main comments are that:

1.      Interest in the filed would be higher for a wider definition for “BRCAness” instead just the BRCA 1&2 genes. I would like to see genes related to the DNA repair pathway of homologous recombination included in the analysis

2.       As the authors mention, the way genetic alterations are measured an/or “BRCAness” is measured/defined alters. This could have been further evaluated and discussed in the manuscript.

Minor comments:

Page 10, lines 343-346 difficult sentence to read.

Figure 3 text size and image resolution too small to read properly

Author Response

Reviewer 1

The authors performed a systematic review and meta-analysis of BRCA gene 1 & 2 genetic alterations in PCa. In general, the manuscript is well written and timely.

Re: We are really grateful to the Reviewer for this comment.

My main comments are that:

  1. Interest in the filed would be higher for a wider definition for “BRCAness” instead just the BRCA 1&2 genes. I would like to see genes related to the DNA repair pathway of homologous recombination included in the analysis

Re: Thanks for the interesting observation. In pages 9-10, lines 314-319 we specify that we focused on evaluating only the rate of BRCA1 and BRCA2 genes, without focusing on other HRD-related gene aberrations which may also serve as novel biomarkers for predicting the efficacy of PARPis. Since the recommendations regarding the HRD test are still preliminary, we studied only BRCA1/2 genes in light of the recent introduction of the BRCA test in clinical practice. We wanted the meta-analysis to have a clinical meaning for everyday clinical practice. In addition, in the articles included in this meta-analysis the study of HRD alterations is quite sparse and not so rigorous like for BRCA mutations. However, we agree with the Reviewer that this topic is surely hot and could be the object of another analysis.

  1. As the authors mention, the way genetic alterations are measured an/or “BRCAness” is measured/defined alters. This could have been further evaluated and discussed in the manuscript.

Re: Thanks for the comment. We explain this topic as a limitation of our study (Page 10, line 369: “Another limitation is the heterogeneity that characterizes the techniques of molecular analysis used in the studies included in the meta-analysis. Different techniques can be different for sensitivity and specificity, and this could contribute to the high heterogeneity found in the incidence of mutations among different studies.”). In the studies included in our meta-analysis, the techniques of molecular analysis are not always explained in detail; therefore, unfortunately, it is not easy to collect such data in a uniform and comparable way.

Minor comments:

  1. Page 10, lines 343-346 difficult sentence to read.

Re: Thanks. We change it and we hope now the sentence is easier to be read.

  1. Figure 3 text size and image resolution too small to read properly

Re: Editor has the original figure. Anyway we add it in the text with the resolution requested by Cancers instructions for authors (300 dpi). The resolution is the same of other figures of the paper.

Reviewer 2 Report

cancers-2264594

Frequency of Germline and Somatic BRCA1 and BRCA2 Mutations in Prostate Cancer: an Updated Systematic Review and Meta-analysis

The entire work is very well written, presents a large number of data relating to mutations in BRCA1 and BRCA2 genes in different types of patients affected by prostate cancer. The authors demonstrated that somatic mutations are more frequent than germline ones and that those in BRCA2 recur more than those in BRCA1. The first conclusion seems a little obvious.

I would insert a paragraph on the limitations of the research.

Author Response

  1. The entire work is very well written, presents a large number of data relating to mutations in BRCA1 and BRCA2 genes in different types of patients affected by prostate cancer. The authors demonstrated that somatic mutations are more frequent than germline ones and that those in BRCA2 recur more than those in BRCA1. The first conclusion seems a little obvious.

Re: we agree with the Reviewer that the first conclusion (somatic mutations more common that germinal mutations) could seem obvious, but actually the magnitude of difference is interesting, because it was not so obvious that the percentage of somatic mutations (7.18%) differs so much from the germline mutations (4.4%). This evidence supports even more the recommendation of the international guidelines to first perform the somatic BRCA testing, limiting the analysis to germline analysis only in cases with tissue not available or analysis non informative. The magnitude of this difference differs from other diseases where the gap between somatic and germinal mutations is not so distant and the first test to be performed can be the germinal one (e.g. breast cancer).

  1. I would insert a paragraph on the limitations of the research.

Re: There is a long paragraph about limitations from line 348 (Page 10) to line 382 (Page 11). If the Reviewer prefers, at Editor’s discretion, we could separate this paragraph from the Discussion, although this is not explicitly requested by the instructions for authors and according to the template of the manuscript.

Round 2

Reviewer 1 Report

Reply by authors fine by me.